# Multiplex Tissue Imaging: Spatial Revelations in the Tumor Microenvironment

**DOI:** 10.3390/cancers14133170

**Published:** 2022-06-28

**Authors:** Stephanie van Dam, Matthijs J. D. Baars, Yvonne Vercoulen

**Affiliations:** 1Center for Molecular Medicine, University Medical Center Utrecht, Utrecht University, 3584 CX Utrecht, The Netherlands; s.vandam@umcutrecht.nl (S.v.D.); m.j.d.baars-8@umcutrecht.nl (M.J.D.B.); 2Oncode Institute, 3521 AL Utrecht, The Netherlands

**Keywords:** tumor microenvironment, multiplex imaging, spatial analysis, PhenoImager, PhenoCycler, MIBI, MALDI-MSI, imaging mass cytometry, single-cell data analysis, cancer

## Abstract

**Simple Summary:**

Cancer is the leading cause of death worldwide, and the overall aging of the population results in an increased risk of a cancer diagnosis during a person’s lifetime. Diagnosis and treatment at an early stage will typically increase the chances of survival. Tumors can develop therapy resistance, and it is difficult to predict how individual patients will respond to therapy. Most studies that aim to resolve this problem have focused on studying the composition and characteristics of dissociated tumors, while ignoring the role of cell localization and interactions within the tumor microenvironment. In the past decade, technological innovations have enabled multiplex imaging analyses of intact tumors to study localization and interaction parameters, which can be used as biomarkers, or can be correlated with treatment responses and clinical outcomes.

**Abstract:**

The tumor microenvironment is a complex ecosystem containing various cell types, such as immune cells, fibroblasts, and endothelial cells, which interact with the tumor cells. In recent decades, the cancer research field has gained insight into the cellular subtypes that are involved in tumor microenvironment heterogeneity. Moreover, it has become evident that cellular interactions in the tumor microenvironment can either promote or inhibit tumor development, progression, and drug resistance, depending on the context. Multiplex spatial analysis methods have recently been developed; these have offered insight into how cellular crosstalk dynamics and heterogeneity affect cancer prognoses and responses to treatment. Multiplex (imaging) technologies and computational analysis methods allow for the spatial visualization and quantification of cell–cell interactions and properties. These technological advances allow for the discovery of cellular interactions within the tumor microenvironment and provide detailed single-cell information on properties that define cellular behavior. Such analyses give insights into the prognosis and mechanisms of therapy resistance, which is still an urgent problem in the treatment of multiple types of cancer. Here, we provide an overview of multiplex imaging technologies and concepts of downstream analysis methods to investigate cell–cell interactions, how these studies have advanced cancer research, and their potential clinical implications.

## 1. Introduction

Tumors and their microenvironments often comprise complex and heterogeneous tissues made up of multiple cell types, mainly tumor cells, immune cells, fibroblasts, and endothelial cells. Interactions between these heterogeneous subsets in the tumor microenvironment (TME) are required for stepwise tumor evolution and tumor progression [1]. In recent years, insights into the TME have increased rapidly due to the development of single-cell multiplex measurements, such as single-cell transcriptomics and proteomics [2]. Many single-cell proteome and transcriptome studies have been performed on single-cell suspensions from dissociated tumors in order to analyze the heterogeneity of the immune microenvironment and how it relates to therapy responses [3]. For example, multiple single-cell studies, employing RNA sequencing and flow cytometry of isolated tumor infiltrating T cells, have demonstrated that T cell exhaustion in the TME can predict the response to immune checkpoint inhibition [4,5]. These single-cell approaches have led to the discovery of various tumor-specific cell types and activated transcriptional programs in the TME relevant to cancer evolution, cancer progression, or patients’ treatment responses. While these studies bring valuable and novel insights, single-cell isolation can cause the loss of cell types and proteins that are sensitive to dissociation methods, and disregards the extracellular matrix. Moreover, single-cell suspension analyses inevitably overlook tissue organization and specific cell–cell interactions in pathology. To overcome these limitations, spatial analysis technologies have been developed over the last decade to analyze the intact TME, which have, for example, revealed B cell and T cell co-localization in organized tertiary lymphoid structures (TLS) [6]. Some of the multiplex spatial technologies that have been developed utilize fluorescence-based microscopy [7,8,9], and others make use of antibody-targeted sequencing [10] or mass spectrometry-based detection to generate spatial expression data [11,12,13]. Spatial data has opened up new possibilities for studying specific tissue regions and cell subtypes, local cellular behavior marked by active signal transduction or receptor expression, and interactions between diverse cell types in the TME. Here, we review spatial imaging technologies that have been developed for multiplex targeted transcriptome analyses, and both targeted and untargeted proteome analyses. Then, we discuss how these data can be studied using tools in open-access software to explore TME organization and heterogeneity [14,15,16]. This review highlights insights into cancer biology and prognosis that have been generated in the last decade by applying different spatial multiplex imaging methods and analysis methods, as summarized in Figure 1.

## 2. Discoveries within the TME Using Multiplex Imaging Methods

In recent years, novel technologies have allowed for the development of spatial multiplex imaging, which enables the simultaneous analysis of >5 markers of interest on a tissue slide while conserving the spatial context. Multiplex spatial imaging methods allow the generation of large datasets containing clinically relevant biological data, consisting of multi-layered information. A schematic overview of each discussed method is provided in Figure 2, Figure 3, Figure 4, Figure 5, Figure 6 and Figure 7. Additionally, the technical specifications, advantages, and disadvantages of each multiplex method are summarized in Table 1. These multiplex spatial methods facilitate in-depth investigation of TME heterogeneity within and between various cancer types. Reports of multiplex spatial imaging studies often provide detailed descriptions of differences in cellular composition, novel cellular phenotypes, or co-localization of specific cell types. Observations made by different multiplex spatial imaging methods are summarized in Table 2 and will be discussed below. Finally, data of these multiplex spatial imaging methods are used for tumor classification and grading, clinical outcomes, therapeutic responses to immune checkpoint inhibitors, or other therapies like chemotherapy, radiotherapy, or chemoradiotherapy. All clinically relevant applications are listed per method in Table 3.

### 2.1. Development of the Imaging Field

The first key studies used immunohistochemistry to analyze a limited number of markers simultaneously [19]. Immunohistochemistry has contributed to the development of the ‘immunoscore’, which showed for the first time that, compared to histopathological analysis, immunological imaging data of T cells in tumors provide a better predictor of the clinical outcome in patients with colorectal cancer (CRC). Besides immunohistochemistry, multiplex immunofluorescence is routinely used to stain the TME. Multiplex immunofluorescence is a method combining one to five different antibodies coupled with fluorophores. This method allowed for the first identification of distinct cellular phenotypes within the TME and is routinely used in diagnostics. The advantage of this method is that fluorophores can be imaged simultaneously, and the signal has a large linear range which allows quantification. These fluorophores must be chosen carefully to prevent signal spillover, and fluorophore combinations are limited.

### 2.2. PhenoImager

To minimize spillover in other channels, a multispectral imaging approach is required. A decade ago, the multispectral imager, PhenoImager, formerly known as Vectra (Figure 2), became commercially available to study the TME. This method allows the use of four to seven primary antibodies plus one nuclear counterstain. The antibody staining is performed in repetitive cycles of one primary antibody, secondary horseradish peroxidase (HRP)-labeled antibody, followed by addition of an opal polymer. This HRP converts the opal fluorophore with a short half-life. The primary and secondary antibodies are stripped, the following staining cycle begins, and finally the tissue slide is imaged using the PhenoImager. This method has been used to study tumor heterogeneity [41], cell–cell interactions [28], the spatial localization of subsets [9,29,45], prognosis prediction, [33,34,46,47,48], evaluation of therapeutic responses to immune checkpoint inhibition (ICPI), and changes in the immune landscape after chemotherapy with or without radiotherapy [49,63,64,72].

**Figure 2 cancers-14-03170-f002:**
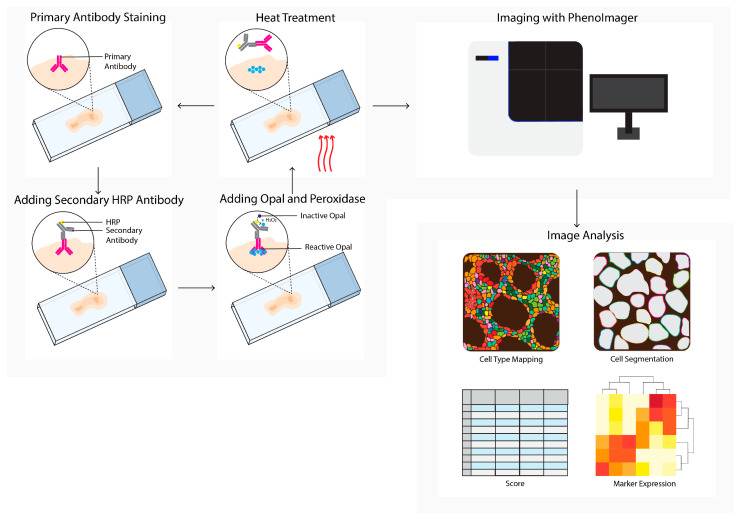
PhenoImager (Vectra) workflow. PhenoImager allows for the use of up to six primary antibodies (or eight in case of the high throughput version) and a nuclear stain. Antibody staining is performed in repetitive cycles of one primary antibody, a secondary horseradish peroxidase (HRP)-labeled antibody, followed by the addition of an opal polymer. HRP converts the opal fluorophore when peroxidase is present. Next, the primary and secondary antibodies are stripped by heat treatment, followed by the next staining cycle, and finally the tissue slide is analyzed using the PhenoImager microscopy system, resulting in data images. Cells in these images can be segmented and downstream analysis can be performed (e.g., cell type mapping and marker expression).

#### 2.2.1. Cell–Cell Interactions and Spatial Localization within the TME

The first study used the Vectra imager to investigate spatial interactions between T cells and tumor cells based on the cross-type nearest neighbor distance distribution function (G-cross) in 120 samples of non-small lung cancer (NSCLC) patients [28]. Spatial data analysis revealed that interactions between regulatory T cells (T_reg_) and tumor cells in the tumor core, but not in the invasive margin, were associated with overall survival. Moreover, interactions between CD8^+^ T cells and T_reg_ within the invasive margin were associated with overall survival, but not when these interactions were in the tumor core.

Another study characterized the cancer-associated fibroblasts (CAFs) in the invasive margin of thirty tumors (stage III) of CRC patients [9]. Stoma-high colorectal tumors showed a reduced number of specific CAFs (FSP1^+^CD45^+^ cells) and increased fibroblast activation protein (FAP) expression, compared to stoma-low tumors. These FAP-expressing CAFs were most abundant in the invasive margin, compared to the tumor center, in stroma-high colorectal tumors. Together, these studies demonstrated that specific cellular localization in the tumor and cellular interactions can distinguish tumor subtypes and/or predict clinical outcomes.

#### 2.2.2. Clinical Data Correlated to Spatial Imaging Analysis

Next, different research groups explored the correlation between immune cell spatial localization and clinical outcomes for various treatments, such as platinum-based chemotherapy, chemoradiotherapy, immunotherapy, or vaccination in combination with ICPI. For example, the immune cell composition in twenty-four human papillomavirus 16 (HPV16), positive solid tumors, from patients treated with HPV16 peptide vaccine in combination with twelve months of nivolumab, were analyzed [49]. The total amount of macrophages (PD-L1^+^CD68^+^, PD-L1^−^CD68^+^), both activated cytotoxic T cells (CTLs) (PD-1^+^CD8^+^CD3^+^) and T cells (PD-1^+^CD3^+^), was only increased in responders. Notably, only the expression of the checkpoint C3a receptor (C3aR) on macrophages correlated with progression-free survival.

Another study investigated the treatment responses of seventy-five patients with rectal cancer who received preoperative chemoradiotherapy [72]. Treatment responses were divided into two groups of patients with either total regression, or a minimal, moderate, or near-total response. Here, mean CD4^+^ T cell infiltration and PD-L1^+^ lymphocyte density, as well as the ratios of CD4:PD-L1, CD8:PD-L1, and FOXP3:PD-L1, increased significantly in rectal tumors, which showed total regression.

Two other studies also captured the changes in T cells in the TME upon treatment. The first study used nine paired pre-treatment and post-treatment samples of patients with high-serous stage III-IV ovarian carcinoma, treated with carboplatin and taxane (paclitaxel or docetaxel). Before treatment, tumors contained high counts of stromal macrophages, the numbers of which were reduced after therapy [63]. Moreover, in the majority of the patients, T helper cell (T_h_ cell):T_reg_ cell ratios changed, shifting towards a balance in which the T_h_ cells outnumbered the T_reg_ cells. Furthermore, the proximity of ICOS^+^ T_h_ cells and T_reg_ cells decreased post-treatment.

The second study included twenty-two paired pre-treatment and post-treatment immunotherapy-treated samples from patients with stage III-IV head and neck squamous cell carcinoma (HNSCC). In this cohort, seven patients with a good therapy response showed no significant differences in CD3^+^ and CD4^+^ T cell numbers compared to patients with poor treatment response [64]. Patients with a good therapy response showed significantly increased CD8^+^ T cell numbers three to four weeks after cetuximab treatment. Collectively, these studies show that specific immune cell subsets, cellular ratios, and checkpoint expressions could change upon receipt of different treatments, which could be related to responses to specific therapies.

### 2.3. PhenoCycler

PhenoCycler, formerly known as CO-Detection by Indexing (CODEX), is a commercially available multiplex tissue imaging platform that uses oligonucleotide conjugated antibodies to detect up to one hundred markers in the same tissue (Figure 3). Initial antibody staining is followed by hybridization cycles with three reporter oligonucleotides containing spectrum-separable fluorophores. These reporter oligonucleotides hybridize with the unique antisense oligonucleotide that is conjugated to the primary antibody. After each cycle, a microscopy image is acquired, followed by removing the fluorophores and a novel hybridization cycle. Upon the completion of all cycles, the images are compiled and aligned to achieve a multiplex image [7]. This cycle-dependent approach drastically increased the number of potential markers, and opened opportunities for detailed characterization and spatial exploration of the TME [20,42]. 

**Figure 3 cancers-14-03170-f003:**
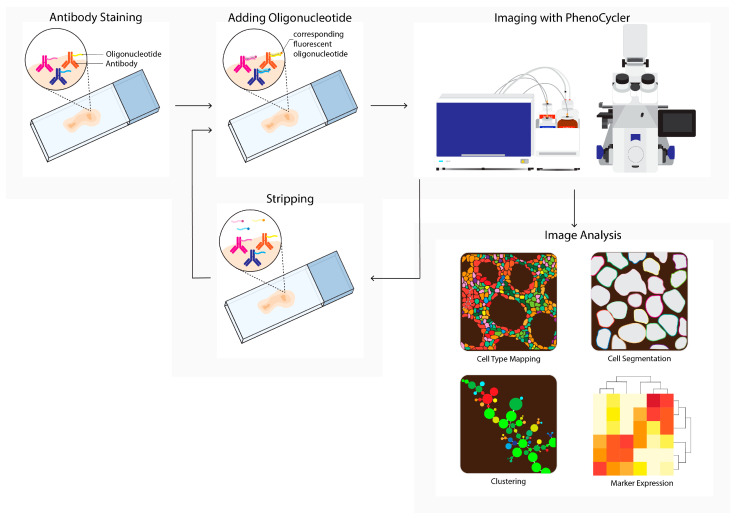
PhenoCycler (CODEX) workflow. Tissue is labeled using oligonucleotide-conjugated antibodies to detect up to one hundred markers simultaneously. Initial antibody staining is followed by hybridization cycles with three reporter oligonucleotides containing spectrum-separable fluorophores, which hybridize with the unique antisense oligonucleotide conjugated to the primary antibody (left). After each cycle, a microscopy image is acquired (imaging with PhenoCycler). Next, reporters are removed; this is followed by the next reporter hybridization cycle. Images from all cycles are compiled and registered to generate multiplex data images. During imaging analysis, these images can be processed into single-cell expression data and downstream analysis is performed (e.g., cell type mapping, clustering, marker expression).

#### Characterization and Spatial Exploration of the TME

One of the first studies to explore the spatial characterizations of the TME using PhenoCycler was performed in a stage III-IV CRC cohort. This study employed a fifty-six-marker panel to investigate differences between thirty-five colorectal tumors with de novo TLS at the invasive front, compared to tumors that lacked TLSs but contained a diffuse tumor infiltrate [42]. The single-cell data were visualized as cellular neighborhoods and revealed certain differences. For example, cellular neighborhoods enriched for granulocytes and PD-1^+^CD4^+^ T cells were positively correlated with overall survival for patients with colorectal tumors that displayed diffuse infiltrates. Moreover, overall survival in these patients was associated with cellular neighborhoods containing high CD4^+^ T cells frequencies and CD4^+^/CD8^+^ T cell ratios. In contrast, frequencies outside these cellular neighborhoods did not influence the overall survival rate.

More recently, PhenoCycler has also been used to elucidate the role of the S100A7/cytosolic phospholipase A2 (cPLA2)/Prostaglandin E2 (PGE2) signaling pathway in breast cancer; it was found that S100A7 is associated with tumor growth and metastasis [20]. Breast cancer-bearing mice that overexpressed S100A7 were treated with the cPLA2 inhibitor, and the analysis revealed changes in T cell composition and cellular interactions. Cluster analysis revealed that cPLA2 inhibitor treatment promoted infiltration of activated CD4^+^ T cells and CD8^+^ T cells within the tumor. Furthermore, the highest degree of interaction among CD4^+^ and CD8^+^ T subsets was observed after cPLA2 inhibitor treatment. Studying these anti-tumor responses in relation to tumor-infiltrated T cells during treatment is essential for establishing whether the number of infiltrating immune cells is correlated with the anti-tumor response. Additionally, studies that focus on immunological changes within the TME will help to uncover biological mechanisms induced by the treatment, and enable biomarker discovery to predict clinical responses.

### 2.4. Multiplexed Ion Beam Imaging by Time-of-Flight

Multiplexed ion beam imaging by time-of-flight (MIBI-TOF, Figure 4) can theoretically utilize up to 100 primary antibodies, each of which can be coupled to unique metal isotopes. A multiplex antibody panel is incubated on a tissue section, and a primary oxygen duoplasmatron ion beam applies a raster to the sample surface. Next, the beam ionizes the metal-conjugated antibody-containing rasters, which are subsequently detected as secondary ions separated by mass and charge by time-of-flight mass spectrometry. The mass spectrometry data are integrated with the single-cell images to generate a tabular chart that contains single-cell data, which can be used to generate spatial profiles [12]. MIBI-TOF is used to study tumor spatial heterogeneity [17,21,35] and tumor progression [80].

**Figure 4 cancers-14-03170-f004:**
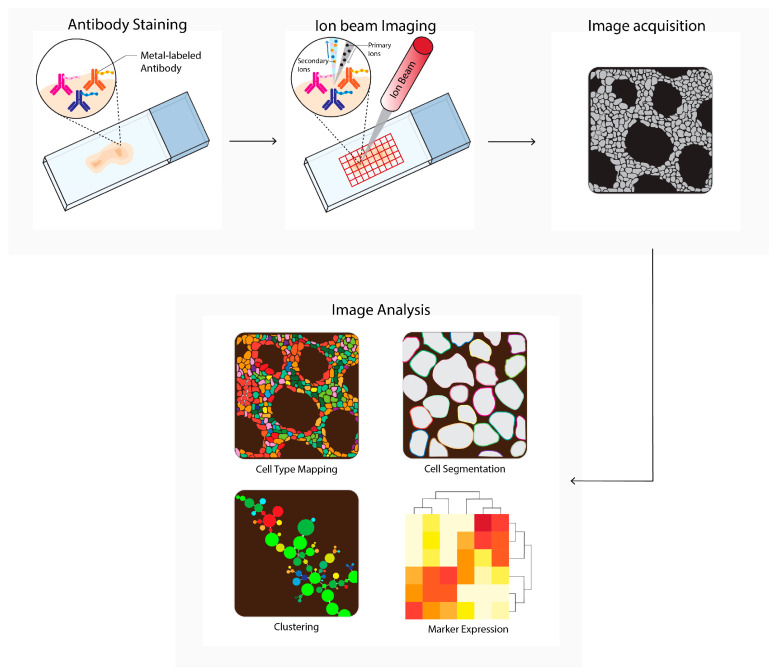
Multiplexed ion beam imaging by time-of-flight (MIBI-TOF) workflow. Tissues are first labeled with a multiplex panel of antibodies conjugated with heavy metals containing polymers. Next, these are directly ionized to generate secondary ions. Ions are filtered and detected by time-of flight mass spectrometry. Next, multiplex images are generated, containing images depicting expressions for each separate antibody–metal conjugate. These images can be processed into single-cell expression data. During imaging analysis, these images can be processed into single-cell expression data and downstream analysis is performed (e.g., cell type mapping, clustering, marker expression).

#### 2.4.1. Tumor Spatial Heterogeneity

MIBI-TOF has been employed to explore the spatial context of the TME and to observe heterogeneity in both tumor and immune cell numbers and in their localization [17,21,35]. For instance, in a cohort containing tissue samples from forty-one TNBC patients, a 36-plex antibody panel, which included phenotypic markers and markers for intratumoral immune cells, was used to investigate the TME [17,21]. This study revealed that the numbers of infiltrating immune cells correlated with the numbers of CD31^+^ vascular endothelium cells [17]. Additionally, patients either showed high CD4^+^ T cell numbers and low macrophage numbers, or the inverse, and most patients displayed enrichment of neutrophils and depletion of B cells.

In a follow-up study, this research group used the same sample set to further analyze the immune cell spatial organization and composition within the TME [21]. Most tumors showed predominantly PD-1^+^CD4^+^ T cells, and follow-up analysis revealed that PD-1 was mainly expressed by CD4^+^ T cells, while the majority of PD-L1 is expressed by macrophages. Spatial analysis revealed that the PD-1^+^CD4^+^ T cells colocalized with CD8^+^ T cells or PD-L1^+^ macrophages in separate immune cell regions. In contrast to the first study, the later analysis showed that B cells were infiltrated in the tumor core instead being depleted. The TNBCs were classified as either compartmentalized, meaning that there are regions predominately composed of immune cells or tumor cells, or as mixed tumors, meaning high spatial mixing of immune cells and tumor cells. Compartmentalized TNBC tumors were associated with PD-1 expression on CD4^+^ T cells and PD-L1 expression on myeloid-derived cells in the tumor–immune cell boarder. In contrast, mixed tumors have PD-L1 expression on tumor cells and PD-1 expression on CD8^+^ T cells. Additionally, the spatial proximity for receptor–ligand pairs was analyzed. This analysis revealed that myeloid-derived suppressor cells are in near proximity of a cluster of tumor cells and lymphocytes. Upon closer image inspection, a concentric cellular organization was observed, consisting of tumor cells aligned with myeloid-derived suppressor cells, followed by a lymphocyte zone.

MIBI-TOF has been combined with other high-plex technologies, spatial transcriptomics, and single-cell sequencing to explore a tumor-specific keratinocyte population in ten cutaneous squamous cell carcinoma (cSCC) tumors [35]. Droplet-based single-cell RNA sequencing of ten paired tumor–skin regions of dissociated tumors was analyzed, and revealed four keratinocyte clusters. Three clusters were present in both the tumor and the normal skin; these clusters were identified as basal keratinocytes, cycling keratinocytes, and differentiating keratinocytes. The fourth cluster was only present in cSCC tumors, and was identified as comprising tumor-specific keratinocytes. Spatial transcriptomics also observed this tumor-specific keratinocyte cluster, which is in concordance with the single-cell RNA sequencing results. Analysis of approximately 100 μm from the tumor border revealed an enrichment for transcripts associated with CAFs and endothelial cells. Integration of the single-cell data and spatial transcriptomics data of the tumor border confirmed that tumor-specific keratinocytes reside in a fibrovascular niche located at tumor borders, which are enriched for CAFs and endothelial cells. To further explore the spatial organization of the tumor border, six tumors were analyzed using MIBI. In-depth single-cell analysis of the immune landscape within the tumor border revealed a correlation between macrophages, CD4^+^ T cells, CD8^+^ T cells, and T_reg_ cells. Interestingly, the T_reg_ cells seemed to function in two different niches, since these T cells co-localize in two different clusters: one containing CD4^+^ T cells and macrophages, and the other containing CD8^+^ T cells.

#### 2.4.2. Tumor Progression

Besides describing different tumor and immune cell compositions, imaging studies have generated insight into specific TME changes that are relevant to tumor progression. To explore which changes in the TME are essential in transitioning from carcinoma in situ into a carcinoma, longitudinal analysis of tissue samples from normal breast tissue that progressed into ductal carcinoma in situ (DCIS) and invasive breast cancer (IBC) was performed in nine patients [80]. The first transition, from normal breast tissue to DCIS, showed a reduction in normal fibroblasts, resting fibroblasts, and mast cells in the tumor stroma. The second transition, from DCIS to IBC, showed that fibroblasts were supplanted by CAFs. Additional spatial analysis searched for risk factors for DCIS to IBC progression. Components that were identified as making a minor contribution to the progression to IBC included: periductal immune cells, and antigen presenting cells that are located nearby PD-1^+^ fibroblasts. Additionally, expression of myoepithelial E-cadherin (MEC) was identified as major risk factor for progressing towards IBC. Moreover, the myoepithelial lining of non-progressors was thinner and discontinuous, whereas the myoepithelial lining of progressors was thicker and continuous; these are features that are also observed in normal breast tissue. Therefore, it was concluded that the loss of normal-like features such as continuous and thicker myoepithelial cells could have a protective function in preventing IBC progression in non-progressors.

### 2.5. Imaging Mass Cytometry

Imaging mass cytometry (IMC, Figure 5) uses a pulsed ultraviolet laser to ablate tissue labeled with metal-labeled antibodies from a glass slide [13]. The ablated tissue is taken up into an argon gas flow and ionized by the plasma, forming ion clouds. These ion clouds pass a filter to enrich the reporter ions and remove common biological elements, after which the signal is quantified by time-of-flight mass spectrometry. The data table integrates single ion signals per pixel, after which image and single cell segmentation can be converted to single-cell data for downstream analysis. In cancer research, IMC is used to discover novel phenotypes and biomarkers, or to study TME heterogeneity [22,23,24,25,30,31,32,36,43,44,50]. IMC is also used to predict prognosis and therapy responses in humans or in mice [23,30,51,65,66,67,68,73,74,75].

**Figure 5 cancers-14-03170-f005:**
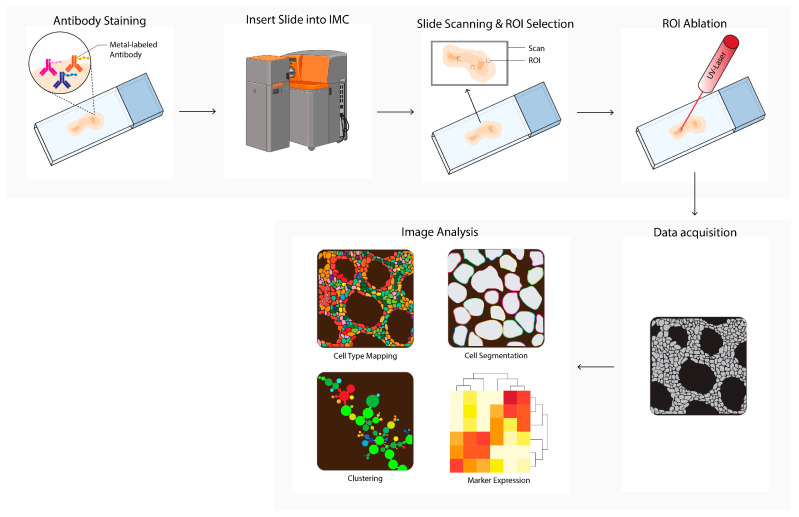
Imaging mass cytometry (IMC) workflow. Tissues are first labeled with a multiplex panel of antibodies conjugated with heavy metal containing polymers. Next, the slide is inserted into the imaging mass cytometer (IMC) and regions of interest (ROI) are selected. Small pieces of 1 uM^2^ of labeled tissue are consecutively ablated with a UV-laser and ionized. Ions are filtered and detected by time-of flight mass spectrometry. Next, multiplex images are generated, containing images depicting expressions for each separate antibody–metal conjugate. During imaging analysis, these images can be processed into single-cell expression data and downstream analysis is performed (e.g., cell type mapping, clustering, marker expression). The Hyperion cartoon was acquired from BioRender.

#### 2.5.1. Prognosis and Therapy Responses

Different studies have explored how the TME is related to disease prognosis or responses to ICPI or platin-based chemotherapy. IMC has been used to study HER2^+^ intracellular domain (ICD) and extracellular domain (ECD) ratios in HER2^+^ breast cancer after adjuvant chemotherapy and subsequent trastuzumab treatment [67]. An elevated ECD/ICD ratio was associated with a lower disease recurrence within five years in trastuzumab-treated patients. Moreover, after treatment, CD8^+^ T cells localized closer to HER2^+^ ECD at the tumor–stroma interface, which decreased from the tumor core to the tumor invasive front. This differential localization pattern may suggest that CD8^+^ T cells play a role in the anti-tumor response after anti-HER2^+^ therapy.

Another study investigated forty-one untreated advanced stage (stage IIIB-IV) high serous ovarian cancer, and compared different immune subsets in long-term survivors (LTS) and short-term survivors (STS) [52]. Tumors of LTS showed an increase in granzyme B^+^ CTLs and significantly higher CD45RO^+^CD4^+^ memory T cells numbers, and a reduction in B7H4^+^Keratin^+^ tumor cells and CD31^+^CD73^mid^ endothelial cells. Analysis of the cell–cell interactions suggested that CTLs deplete tumor cells in LTS, since granzyme B^+^CD8^+^ T cells are negatively correlated with B7H4^+^ tumor cells. Additional nearest neighbor interaction analysis indicated only a prognostic significance in granzyme B^+^CD8^+^ T cells and tumor cell interaction. Overall survival was analyzed using recursive feature elimination (RFE) and showed a positive coefficient of granzyme B^+^CD8^+^ T cells and CD45RO^+^CD4^+^ memory T cells. This result indicates a positive correlation with overall survival.

Finally, ten gastric cancer biopsies of patients treated six months preoperatively with modified folinic acid, fluorouracil, and oxaliplatin (mFOLFOX) were analyzed, and showed that biopsies of patients with a pathological response had higher platinum levels compared to biopsies of non-responders [53].

Besides platin-based treatment responses, IMC has been used to study ICPI responses in various patient cohorts. For example, the immune cell compositions in thirty-five patients with resectable (stage I) and advanced (stage IV) NSCLC, who were treated with a single anti-PD-1 agent, were studied [54]. Patients with advanced stage NSCLC had a higher abundance of burned-out effector (Ebo) CD8^+^ tumor-infiltrating lymphocytes (TIL). Furthermore, patients with non-durable benefits from the anti-PD-1 treatment had significantly higher Ebo CD8^+^ TIL proportions, which showed elevated expression of PD-1, lymphocyte-activation gene 3 (LAG3), and Ki67, compared to patients with long durable benefits from anti-PD-1 treatment. Overall, Ebo CD8^+^ TIL abundance was an independent outcome predictor, which was associated with a poor overall outcome.

#### 2.5.2. Discovery of Novel Phenotypes in the TME

Imaging strategies with multiplex antibody panels enable the discovery of novel (immune) phenotypes in the TME: for example, the discovery of one new immune subpopulation, identified as CD3^−^CD4^+^, in both the tumor region and the adjacent tumor area in primary squamous cell lung carcinomas that did not receive any therapy [24]. Interestingly, these CD3^−^CD4^+^ immune cells expressed forkhead box P3 (FOXP3), terminal deoxynucleotidyl transferase (TdT), and tumor necrosis factor alpha (TNFα), but lacked expression of CD25, CD127, and interferon gamma (IFNγ). Collectively, the expression pattern of these cells pointed towards a proinflammatory function within the TME.

Another example is the discovery of a novel CD4^+^LAG3^+^ T cell subset in classical Hodgkin lymphoma (cHL) using a 35-plex IMC panel [44]. Additional spatial analysis revealed that this specific T cell subset was most abundant in major histocompatibility complex II (MHC-II) negative cHL, but not present in normal reactive lymph nodes. Lastly, analysis of three colon cancer tissue samples, along with the corresponding tissues located next to the cancer, demonstrated that infiltrating EpCAM^+^PD-L1^+^CD4^+^ T cells with upregulated p38-MAPK-MAPKAPK2 signaling were specifically present in the tumor [44].

#### 2.5.3. Biomarker Discovery

To discover new biomarkers to predict anti-PD-1 treatment responses, two non-adjacent tumor cores of sixty melanoma samples, taken from patients treated with nivolumab or pembrolizumab or a combination of both, were stained with a 25-plex IMC panel. The IMC data were analyzed with AQUA software, which is used to study marker intensity across five different compartments: tumor, stroma, macrophages, T cells, and B cells [22]. The IMC data mainly focused on general T cells markers, and showed that higher CD3 and CD8 expression was associated with progression-free survival and overall survival, but CD4 expression was not. Additional compartment-specific analyses revealed that elevated expression of major histocompatibility complex I (MHC-I), beta-2 microglobulin (B2M), CD8^+^ T cells, and LAG3 in both tumor and stroma was associated with progression-free survival. Meanwhile, only colony stimulating factor 1 receptor (CSF1R) expression in the tumor was associated with progression-free survival. IMC data was validated using independent RNA-sequencing data of melanoma patients treated with immune checkpoint inhibition. This validation specifically focused on MHC-I, B2M, and CSF1R expression, and showed that only B2M, but not MHC-I, was correlated with overall survival.

### 2.6. Matrix-Assisted Laser Desorption Ionization Mass Spectrometry Imaging

Matrix-assisted laser desorption ionization mass spectrometry imaging (MALDI-MSI, Figure 6) is a labeling-free method for two-dimensional and three-dimensional quantitative spatial analysis of the molecular distribution of molecules, such as proteins, lipids, metabolites, and glycans in tissue [18]. The microscope slide containing either formalin-fixed paraffin-embedded (FFPE) or fresh frozen (FF) tissue sections is processed, and a matrix is applied. After matrix deposition, the slide is inserted into the MALDI-MSI, and a laser creates a 10–150 micrometer raster. The laser beam ionizes a spot in each raster, and the ionized analytes from the raster are transferred into the mass spectrometer for compound identification. In parallel, an image is created by combining the data of the spot location with the corresponding measured mass spectrum. In the field of cancer research, MALDI-MSI has been used to analyze drug (metabolite) distribution in the TME, tumor subtyping, tumor grading, or biomarker discovery [55,76,77,78,79,81]. MALDI-MSI is also used to predict tumor progression and clinical outcomes [56,69].

**Figure 6 cancers-14-03170-f006:**
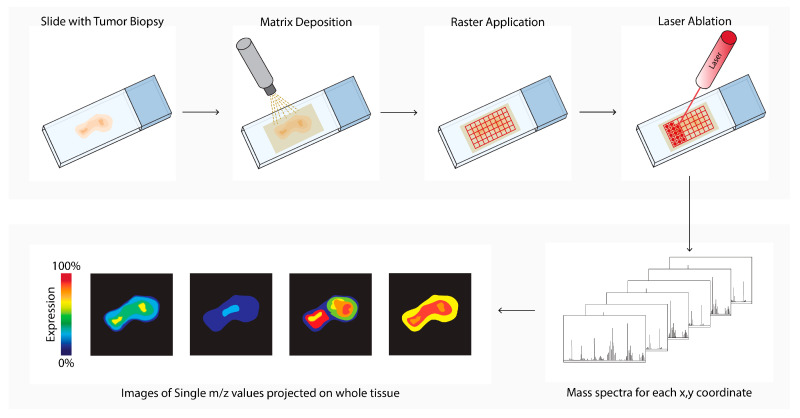
Matrix-assisted laser desorption ionization mass spectrometry imaging (MALDI-MSI) workflow. Tissue sections are processed, and a matrix is applied (matrix deposition). Next, a laser in the MALDI-MSI creates a 10–150 micrometer raster (Raster Application). The laser beam ionizes a spot in each raster (laser ablation), and the ionized analytes from the raster are transferred into the mass spectrometer for compound identification (mass spectra). In parallel, an image is created by combining the data of the spot location with the corresponding measured mass spectrum (images of single *m/z* values).

#### 2.6.1. Tumor (Sub)Classification and Tumor Grading

Non-small cell lung cancer (NSCLC) is the most studied tumor type using MALDI-MSI; this research has revealed differences in the TME that can be used for tumor classification.

For example, elevated collagenase type III levels in the extracellular matrix discriminate low-grade adenocarcinoma (AD) (stage I-III) from adjacent healthy lung tissue. More in-depth analysis revealed that hydroxylated peptides display a specific spatial pattern within the TME, which allows for the differentiation of tumor subtypes [76]. Spatially guided proteomics and histology-guided spatial metabolomics of NSCLC allowed for the differentiation of two NSCLC subtypes: adenocarcinoma and squamous cell carcinoma (SCC). In squamous cell lung carcinoma, glutamine is more abundantly expressed, whereas taurine expression is most abundant in adenocarcinoma [77]. Additionally, spatial protein distribution combined with linear discriminant analysis identification showed that cytokeratin 5/6 (CK5/6), heat shock protein 27 (HSP27), and cytokeratin 15 (CK15) are most commonly (co-)expressed in SCC and, less frequently, in adenocarcinoma. Expression of cytokeratin 7 (CK7) was high in adenocarcinoma, but not in SCC [78].

A similar approach was used to evaluate human epidermal growth factor receptor 2 (HER2) status in pre-defined HER2-status breast cancers. HER2 overexpressing breast cancers showed scattered expression of a molecule with a mass-to-charge ratio (*m/z*) of 8404, while HER2-negative cancers lacked this expression pattern. Next, this molecule was identified using electrospray ionization mass spectrometry as cysteine rich protein 1 (CRP1) [55].

#### 2.6.2. Biomarker Discovery

The results of MALDI-MSI studies have potential implications for tumor (sub)classification and tumor grading, and for the identification of biomarkers with potential prognostic clinical value. For example, nine proteins associated with epidermal growth factor receptor (EGFR) which have potential prognostic value in triple-negative breast cancer (TNBC) were discovered using MALDI-MSI, by comparing tumor tissue to benign breast tissue. In this study, proteins were mapped against a library constructed using liquid chromatography-matrix-assisted laser desorption/ionization mass spectrometry (LC-MALDI-MS/MS), followed by protein network analysis [81].

In NSCLC patients, the proteins neutrophil defensin 1, defensin 2, and defensin 3 were identified as potential markers for predicting responses to anti-PD-1 therapy [79]. In this study, most responding patients and a minority of non-responding patients expressed these proteins in a biopsy taken before treatment. This was validated by immunohistochemical staining, and could be used for the prediction of therapy responses.

#### 2.6.3. Prediction of Tumor Progression and Clinical Outcomes

Besides MALDI-MSI proteome-based discoveries, two independent studies involving patients with CRC stage II or stage III ovarian carcinoma showed that the N-glycan signatures in TME were related to clinical outcomes [56,69].

In CRC, a comparison between tumor cells and adjacent normal colon tissue showed that tumor cells contained reduced levels of highly branched N-glycans and fucosylation, but elevated high-mannose glycan and sialylation levels [56]. Regional comparisons revealed elevated N-glycosylation levels in the tumor cells, which gradually decreased from the tumor cells towards the stroma interface and the tumor lining-stroma. This observation indicates that different tumor and microenvironmental compartments show distinct N-glycosylation patterns. Interestingly, differential glycosylation patterns in the stroma interface that aligns with the tumor cells could distinguish short-surviving from long-surviving patients.

Similar N-glycan distribution effects were detected in the ovarian cancer study [69], demonstrating its ability to discover novel predictive signatures in multiple cancer types. 

### 2.7. Digital Spatial Analysis

Digital spatial analysis (DSP, Figure 7) is a commercially available technique that enables RNA or protein quantification from FFPE or FF tissue by counting targets that are linked to unique indexing oligonucleotides [10]. These indexing oligonucleotides are covalently bound with a UV-photocleavable (PC) linker to a mRNA hybridization probe or to a primary antibody. Tissue slides are first hybridized with mRNA probes or stained with PC-linked primary antibodies. After this first staining, the same slide can be stained with one to four fluorescent labeled antibodies, which allows for the identification of specific tissue structures or cellular subsets of interest. The tissue slide is inserted into the DSP and an image is produced, based on the signal from the fluorescent labeled antibodies. These overview images enable the selection of a region of interest (ROI) of any shape. After the ROI is selected, the digital micromirror device illuminates the ROI, which causes the PC-linked mRNA or primary antibody to be released from the tissue. The photocleaved indexing oligonucleotides are aspirated with a microcapillary and collected in a 96-well plate. The collected photocleaved indexing oligonucleotides are analyzed either by next generation sequencing or by an nCounter system.

**Figure 7 cancers-14-03170-f007:**
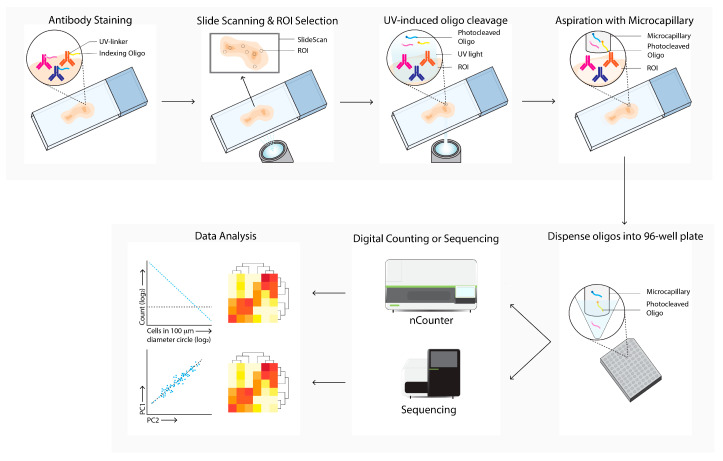
Digital Spatial Profiling (DSP) workflow. Tissue sections are labeled with antibodies and/or in situ hybridization with mRNA probes, which are linked with UV-cleavable oligo-tags. Slides are labeled with fluorescence-conjugated antibodies to determine cell subsets and select regions of interests and masks for bulk cell subsets for directed UV-cleavage of the oligo-tags. The cleaved oligos are collected with a microcapillary and transferred to a 96-well plate. Next, the oligos are quantified by digital counting (nCounter) or next-generation sequencing. Differential expression of specific mRNA or proteins between ROIs and cell subsets are next analyzed (data analysis).

DSP has been used in the field of cancer research to search for TME transcriptional profiling, and to identify predictive markers for prognosis, therapy response prediction, or clinical outcomes [26,57,58,59,60,61,62,70,71,82]. Furthermore, DSP has been utilized to investigate TME heterogeneity [27,37,38,39,40,83,84,85].

#### Immunotherapy Response Prediction

DSP has been widely used to predict immunotherapy responses or to identify prognostic markers. Three different studies have investigated which markers are related to ICPI treatment response.

The first study examined seventeen resectable stage III melanoma baseline biopsies, which were treated with adjuvant or neoadjuvant ipilimumab and nivolumab, before or after complete regional lymph node dissection [71]. Samples were analyzed with twenty-nine oligo-conjugated antibodies directed to immune-related surface antigens. Low expression of CD3, B2M, and PD-L1, and a low IFNγ signature within the tumor were correlated with relapse after adjuvant or neoadjuvant ipilimumab and nivolumab. Meanwhile, tumors with an intermediate or high IFNγ signature did not relapse.

A second study included resectable Melanoma stage III or oligometastatic stage IV tumors that were treated with neoadjuvant nivolumab monotherapy or a combination of nivolumab and ipilimumab [26]. Immunohistochemistry showed that the tumors of responding patients expressed elevated PD-L1 expression in tumor cells, increased lymphocyte marker expression (granzyme B, CD4, FoxP3, CD20, and PD-1), and elevated proportions of CD8^+^ TILs. Furthermore, higher expression of CD45RO, B2M, CD3, CD8, CD19, CD20, and Ki67 in the CD45^+^ immune cell infiltrate correlated with ICPI response. Meanwhile, the T cell receptor sequencing showed that tumors of responders had a larger T cell clone diversity than tumors of non-responders.

The third study investigated which immune cell makers were correlated with ICPI treatment outcomes. Seven samples, taken from patients with metastatic head and neck squamous cell carcinoma that had been treated with nivolumab and pembrolizumab, were analyzed with a 44-marker panel [61]. The immune cell profiling panel revealed that expression of CD4, CD45RO, CD68, IDO-1, P-ERK, Ki67, PD-L1 PD-1, Granzyme B, CD45, OX40, STAT3, P-STAT3, CD44, STING, CD66b, P-AKT, and PTEN was associated with progressive disease. Additionally, in contrast to the second study of melanoma, CD8 expression was not predictive for the ICPI treatment outcome here.

Two studies used immunofluorescent antibodies to identify specific compartments within the TME. The first study included immunotherapy-treated melanoma samples. These samples were stained with a 44-plex antibody cocktail that was mainly directed towards the immune-related surface antigens of leukocytes and macrophages [59]. Additionally, some markers to identify the tumor cells were included. These makers were used to identify three compartments: melanocytes (S100^+^ and HMB45^+^), macrophage (CD68^+^), and leukocyte (CD45^+^) compartments. Analysis of these compartments revealed that high CD8^+^ counts within the macrophage compartment were associated with progression-free survival, as well as immunotherapy response and prolonged overall survival. CD8^+^ numbers within the tumor compartment were associated with favorable outcomes, whereas CD8^+^ numbers in the leukocyte compartment were not. In total, eleven markers and fifteen markers were found to correlate with progression-free survival and overall survival, respectively. However, in the multivariate analysis, only PD-L1 expression in the macrophage compartment was significantly associated with progression-free survival and overall survival.

The second study included eighty-one NSCLC patients (stage III-stage IV) who were treated with either nivolumab, pembrolizumab, or atezolizumab [60]. Three different compartments within these tumors were analyzed: tumor cells (pan keratin^+^), immune cells (CD45^+^), and macrophages (CD68^+^). Multivariate analysis showed that high levels of CD56 and CD4 in the immune cell compartment are predictive of longer progression-free survival, prolonged overall survival, and durable benefit, while high VISTA levels and CD127 levels in the tumor compartment were predictive of non-durable benefit and shorter progression-free survival.

## 3. Challenges and Developments in Spatial Data Analysis

In recent years, the development of multiplex imaging technologies has increased the need for methods to analyze imaging data, particularly spatial single-cell analysis. ImageJ [86] is the major open-source software package used for many different image analysis tasks, including single-cell segmentation, tumor grading of clinical images originating from magnetic resonance imaging (MRI), or staining quantification [87,88,89]. A more recent open-source software package specifically developed for tissue analysis, QuPath [90], is now being applied to multiple spatial tissue analysis tasks, including region annotation, cell segmentation, marker expression, and distance calculation.

### 3.1. Cell Segmentation

A major challenge in tissue analysis is cell identification [91]. Cell identification requires that the boundaries of cells are identified, and pixels are assigned to a specific cell. This process is called single-cell segmentation. Cell segmentation allows marker expression analysis on a single-cell level, and analysis of the localization of each cell. Single-cell segmentation is particularly challenging in dense tissues because cells can overlap in the z-dimension, which, when imaging, is perceived as overlapping membranes or nuclei. Overlap of cellular parts complicates the identification of individual cells for single-cell segmentation. Moreover, tissues often contain a variety of cell types with various shapes. To identify and classify specific (immune) cell subsets, a specific combination of (multiple) membrane, cytoplasmic, and/or nuclear marker expression is required.

For multiplex imaging data, segmentation is mostly performed using machine learning algorithms. These algorithms include user-guided machine learning pipelines with Illastik [92] and CellProfiler [93]. These pipelines have been used in the analysis of datasets that combine IMC and fluorescent microscopy [16,89,91]. More recently, deep-learning algorithms that use convolutional neural networks (CNNs) have become available; these include DeepCell [17,94], U-NET [95], and DeepImageJ [96]. The methods discussed here are open source, can be adapted to address specific questions, and require computational and imaging data analysis experience.

Notably, all machine learning is based on training that uses manual data annotations (ground truth) generated by expert users. To avoid annotation bias, which can affect the resulting cell segmentation, it is recommended that multiple experts annotate the training data [97]. Additionally, interobserver variability can be introduced during single-cell generation when the expert assesses segmentation quality or excludes samples based on visual inspection. This might ultimately lead to statistical interference; however, the field does currently not have a gold standard and relies completely on the experience of experts.

### 3.2. Tissue Organization and Cellular Localization

The recent implementation of high-plex imaging technologies allows for the analysis of many differential cell-subtypes in tissues. On one hand, this has sparked the development of novel methods directed towards cell-subset identification and cellular localization. On the other hand, it allows for the analysis of cell–cell interactions, including neighborhood analysis. These analyses can, for example, be performed with HistoCAT, the Histo-Cytometric Multidimensional Analysis Pipeline (CytoMAP), and Squidpy [15,16,98]. Unbiased analysis can reveal spatial patterns; for example, the enrichment z-scores of receptor–ligand pairs expressed in cells have been used to identify the co-localization of proteins [17]. Similarly, single-cell expression data have been analyzed using Pearson correlation heatmaps in CytoMAP [98]; this analysis shows a correlative expression of proteins in neighboring cells within a chosen radius. Such unbiased analyses can show differential tissue organization, but can also reveal which expressed (immune) markers, signaling molecules, or immune checkpoints either colocalize or avoid each other in specific conditions. Another application of CytoMAP is the identification of cellular neighborhoods, by performing clustering on patterns of the spatial co-occurrence of single-cell marker expression [98].

Cell location and gene expression variation can also be analyzed directly, without cell-type assessment or environmental variables, by using spatial variance component analysis (SVCA) to model the spatial sources of the variable expression of different proteins or genes [99]. The application of SVCA to IMC data from breast cancer samples demonstrated differential contributions across detected protein levels of cell-intrinsic effects, environmental effects, and cell–cell interactions. SVCA analysis per protein marker revealed that cell–cell interaction, and environmental and intrinsic effects each contribute differently to the variable expression of specific proteins. It was shown that CD44, CD20, CD3, and CD68 demonstrated the largest cell–cell interaction effects. Additionally, the spatial variance signature aligned with the tumor grade: specifically, the spatial variance signatures of a subset of proteins, including CD44 and CD20 [99].

Deep learning can also be applied here to study spatial patterns. NaroNet is a deep learning framework that combines multiplex imaging and the corresponding clinical patient parameters to perform patch contrastive learning [100]. Patch contrastive learning divides images into patches, which are embedded into a 256-dementional vector. Similar vectors are then assigned by an unsupervised trained CNN to patches that contain a similar biological structure. An enriched graph of these patches is created, which contains spatial interactions between the patches. The enriched graph is the input for NaroNet, which uses the data to classify patients based on the abundance of cell phenotypes, phenotype neighborhoods, and the area of interaction of those neighborhoods within the TME.

NaroNet has been applied to an endometrial carcinoma dataset which included 336 seven-color images from twelve patients with high-grade endometrial carcinoma. Here, four patient labels were added to the data: somatic DNA polymerase epsilon (POLE) mutation status, copy number variation, DNA mismatch repair deficiency status, and two tumor histopathology types. NaroNet identified a specific neighborhood containing two phenotypes (T cells and tumor cells) that were associated with somatic POLE mutation. Moreover, this study used a set of 382 images from 215 breast cancer patients, which were stained with a 37-plex IMC panel [23,100]. Patients were clustered into three risk groups based on their overall survival. NaroNet was able to predict the patient risk group based on 57 learned distinct spatial patterns. In-depth analysis of differences in the TME composition revealed two neighborhoods that were predictive of the distinctions between high-risk and low-risk patients. The first neighborhood contained tumor cells and the other contained fibroblasts, which were more abundant in high-risk patients.

More directive (biased) analysis approaches range from cell–cell interaction analyses to distance calculation between specific cell types, or between a cell type and a tissue region. This type of analysis can reveal differential interactions, distancing between cells, or regional separation [17,98]. Such analyses can, for example, reveal differential tumor archetypes based on lymphocyte infiltrate: tumors contain no infiltrate (cold tumors), infiltrate between tumor cells (mixed tumors) or infiltrate that separates the neighboring compartments (compartmentalized tumors) [17].

### 3.3. Data Management

One of the other challenges of high-plex spatial imaging concerns imaging data management and metadata management [101].

#### 3.3.1. Metadata

Metadata are generated at multiple stages during these high-plex spatial imaging experiments [101]. First, tissue is sectioned and prepared for staining, and clinical data are then associated with these tissue samples, generating clinical and biospecimen metadata. Next, samples are stained with an antibody cocktail and data are acquired using the imaging instrument of choice, which generates channel-level metadata and instrumental metadata, respectively. All of these forms of metadata collectively form the experimental metadata, and, to promote interpretation and data reuse, there is an urgent need to store such metadata in line with the Findable, Accessible, Interoperable, and Reusable (FAIR) standards [102]. Additionally, researchers need to account for the fact that patient data need to be stored anonymously, and the metadata must not allow specific patients to be traced or identified [103].

Recently, a paper highlighted the urgent need for metadata and data standards for high-plex spatial data [101]. One of the key challenges is to implement a metadata standard that balances easy data entry, sufficient detailed information, and reproducible analysis and publication. However, a standard for these high-plex spatial imaging techniques has not yet crystalized, and needs to be further developed by the research community in the coming years.

#### 3.3.2. Data Storage and Distribution

After data acquisition, the data must be stored, processed, and analyzed, and the challenge of data management arises. Both raw data and data generated during processing, up to the point where the final figures are created, must be stored in an organized manner.

One of the challenges in data management concerns the large size of raw multiplex imaging datasets, which can easily require a few terabytes of disk space, a requirement that increases further during data processing and analysis. This makes local data storage and distribution of these large FAIR datasets challenging and costly. Another challenge relates to the question of which different levels of data should be made publicly available from different data analysis stages upon publication. [101]. Recently, a paper proposed that from the quality controlled, assembled images and onwards should be made publicly available, but this is not yet a gold standard. The distribution and storage of such FAIR datasets in a standardized structure, containing multiple data levels, will need to evolve over the coming years to promote accessibility and reproducibility in the near future [101].

## 4. Conclusions and Future Outlook

Recent advances in spatial imaging technologies allow for the study of tissue compartmentalization, cellular localization, and cell–cell interactions within the TME. At the same time, a large marker set allows relatively deep subset identification. Together, these advances reveal potential correlations between therapeutic responses and clinical outcomes. These advances create rich spatial datasets, which also drives the need for computational analysis methods and pipelines for analysis. This allows for faster and deeper gains in biological insights into factors including biomarkers for tumor subtyping, and patient prognosis and/or response prediction. In this review, we surveyed the recent contributions that each single multiplex imaging method has made towards our current understanding of the TME, and their potential clinical implications, along with current methods for single-cell and spatial analysis.

The emergence of the multiplex imaging methods discussed here increased the need for advanced analysis pipelines to generate single-cell data. Before these single-cell data are generated, there are many challenges and potential pitfalls that need to be addressed. The current challenges include normalization, the correction of batch effects, and the removal of experimental artefacts, for which no standardized methods nor a gold standard currently exists. Moreover, (deep) machine learning is often incorporated into the analysis pipeline, and training the algorithm requires many manual annotations. After adding all of the annotations, the predicted segmentation requires a visual inspection to verify its quality. This visual inspection can be challenging due to dataset size and the computation capacity that is required to generate all of the images in parallel. Another challenge in generating single-cell data from cell-dense areas arises from overlapping cells which can be segmented in too many fragments, since pixels of two different membranes or nuclei overlap [104]. This risk of segmentation in too many fragments is increasing when the imaging data has a lower resolution. To improve single-cell segmentation quality in dense areas in lower resolution images, the MATISSE pipeline has recently been released [89,91]. A final challenge that is faced in single-cell data generation is the segmentation of irregularly shaped cell types, such as macrophages. The segmentation of these immune cells requires membrane markers to indicate the irregularly shaped cell boundary. Therefore, segmentation based on nuclear expansion, which assumes a regularly shaped cell boundary surrounding the nucleus, does not perform properly.

The field of multiplex imaging and cancer research will likely start combining and integrating multiple spatial (omics) techniques in the coming years. The first step towards simultaneously quantifying DNA, RNA, and protein levels within the TME was recently taken, by developing Protein Additionally Nucleic Acid In Situ Imaging (PANINI) and coupling it to MIBI [105]. Because of the rapid development of these multiplex imaging techniques, single-cell analysis pipelines can be expected to further improve segmentation quality; moreover, this research field will develop gold standards for normalization, batch correction, and experimental artifact correction. Along with the expanding analysis quality, the existing descriptive research into the TME will evolve towards research that contains comprehensive integration of both clinical parameters and biological parameters. The integration of these parameters will even allow for an improved prediction of biomarkers, clinical responses, and prognosis of cancer patients in the future.

## Figures and Tables

**Figure 1 cancers-14-03170-f001:**
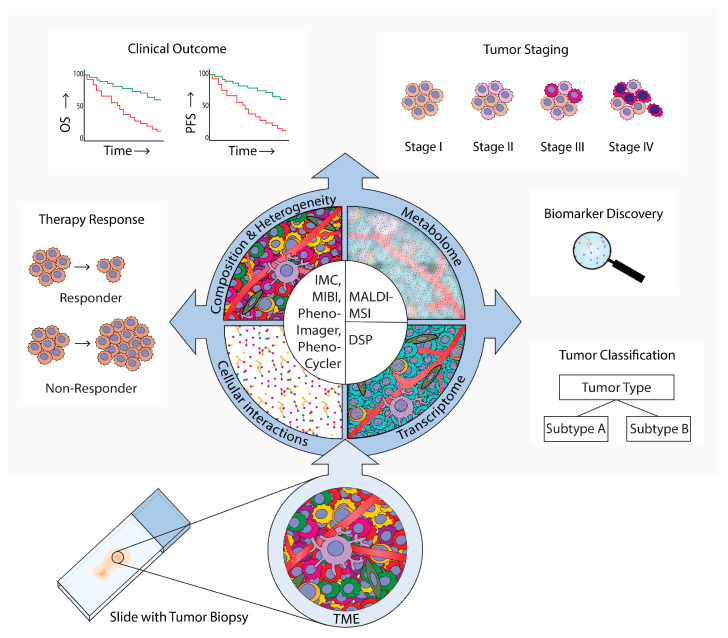
Spatial analysis of the tumor microenvironment (TME). Methods, results, and implications in cancer research. This review provides an overview of technologies that are used for TME spatial analyses in cancer research. These technologies employ microscopy (PhenoImager, PhenoCycler), mass-spectrometry (imaging mass cytometry (IMC)), multiplexed ion beam imaging by time-of-flight (MIBI), and (matrix-assisted laser desorption ionization mass spectrometry imaging (MALDI-MS)), or digital spatial profiling (DSP). PhenoImager, PhenoCycler, IMC, and MIBI can be used to investigate single cells and to explore the cellular composition, heterogeneity, and cellular interactions of the TME. MALDI-MSI can be employed to investigate the TME metabolome in specific regions, and DSP can be used to explore the transcriptome in bulk cells of up to three different subsets. These multiplex spatial methods have provided novel insights into specific biomarkers and TME spatial hallmarks that can be used for tumor subtype classification. Moreover, these methods have uncovered which TME characteristics are related to tumor evolution and progression to advanced stages, clinical prognosis parameters such as overall survival (OS) and progression-free survival (PFS), and prediction of therapy responses.

**Table 1 cancers-14-03170-t001:** Summary of technical details, advantages, and disadvantages of each multiplex imaging method.

Technique	Ref.	Principle	Multiplex	Tissue	Applications	Resolution *	Advantages	Disadvantages
Targeted Multiplex Imaging Approaches Using Antibodies								
Pheno- Imager	[9]	Spectral immuno- fluorescence	Up to six fluorophores (+one nuclear counterstain) or eight fluorophores (+one nuclear counterstain) using the high throughput version	FFPE, FF	TME phenotyping, prognosis and therapy response prediction	Adjustable, max 200 nm	- Allows imaging of large tumor areas - No spillover - One round of imaging - Non-destructive - Automated or semi-automated - Adjustable resolution	- Requires PhenoImager system - Requires pre-designed or self-coupled antibodies
Pheno- Cycler	[7]	Cyclic staining with oligo-nucleotide-conjugated antibodies	≈66 makers, depending on the number of cycles **	FFPE, FF	TME phenotyping, prognosis and therapy response prediction	Adjustable, max 200 nm	- Allows imaging of large tumor areas - Automated or semi-automated assay - Non-destructive - Allows generation of single cell data - Adjustable resolution	- Requires Phenocycler Fusion system - Multiple cycles of imaging of the same area - Requires pre-designed or self-coupled antibodies - Throughput hours to days depending on the cycles
MIBI-TOF	[12]	Metal- labeled antibodies	Up to 40 markers **	FFPE, FF	TME phenotyping, prognosis and therapy response prediction	Adjustable, max 260 nm ^#^ [17]	- Non-destructive - Adjustable resolution - Allows generation of single cell data	- Requires a specific machine - Non-automated or semi-automated assay - Requires pre-designed or self-coupled antibodies
IMC	[13]	Metal- labeled antibodies	Up to 40 markers **	FFPE, FF	TME phenotyping, prognosis and therapy response prediction	1000 nm	- No spillover - One round of imaging - Allows generation of single cell data	- Requires a specific machine - Limited resolution (1000 nm) - ROI selection - Requires pre-designed or self-coupled antibodies - Destructive - Non- or semi-automated
DSP	[10]	PC-linked mRNA hybridization probe or primary antibody	Up to 800 targets	FFPE, FF	TME phenotyping, prognosis and therapy response prediction	10 μm	- Non-destructive - Allows generation of single cell data - Conserves spatial transcriptome data	- Requires a specific machine - ROI selection - Requires pre-designed or self-coupled antibodies
**Untargeted Multiplex Imaging**								
MALDI- MSI	[18]	Labeling-free technique used for (relative and absolute) quantitative and spatial analysis of the distribution of molecules	Global identification of thousands of biomolecules	FFPE, FF	Tumor subtyping, tumor grading, biomarker discovery for therapeutic response or prognosis prediction	600 nm	- Identify unknown biomolecules (glycans, proteins, lipids, and metabolites) - Does not require antibody labeling - High sensitivity and specificity	- requires a specific machine - limited resolution - ROI selection - Destructive

* This includes the theoretical maximum resolution. ^#^ A resolution limit of 260 nm is mentioned in a recent publication; however, the data are acquired with a resolution of 500 nm. ** Theoretically, there is no upper limit, but published data currently show a limit of 40 markers for MIBI-TOF and IMC, mainly due to reagent availability. Published data for the PhenoCycler currently show a limit of 66 markers. Abbreviations: DSP: digital spatial analysis, FF: fresh frozen, FFPE: formalin-fixed paraffin-embedded, IMC: imaging mass cytometry, MALDI-MSI: matrix-assisted laser desorption ionization mass spectrometry imaging, MIBI-TOF: multiplexed ion beam imaging by time-of-flight, PC: photocleavable, ROI: region of interest, TME: tumor microenvironment.

**Table 2 cancers-14-03170-t002:** Summary table of multiplex imaging studies that describe tumor microenvironment heterogeneity, specific cell types, novel immune cell subsets, cellular interactions, or (disease) comparisons.

Category	Method	Ref	Year	Cancer Type	Described Observations
**TME** **Heterogeneity**	Pheno- Imager	[19]	2020	Breast and lung cancer	Different (immune) cell compositions within the TME
	MIBI-TOF	[20]	2019	Breast cancer	Immune cell subset balances and compartmentalization within TNBC TME
	MIBI-TOF	[21]	2018	Breast cancer	Different immune cell compositions and immune cell subset balance within the TNBC TME
	IMC	[22]	2020	Breast cancer	Various (immune) cell compositions within the TNBC TME
	IMC	[23]	2021	Lung cancer	Various (immune) cell compositions within the NSCLC (SCC) TME
	IMC	[24]	2021	OSCC	Various (immune) cell compositions within the TME
	IMC	[25]	2021	Bladder cancer	Different immune cell compositions within the TME
	DSP	[26]	2019	Prostate cancer	Different (immune) cell compositions and signaling pathways within the TME of lytic and blastic bone metastasis
	DSP	[27]	2021	Prostate cancer	Inter- and intra-tumoral differences in (immune) cell compositions in metastatic prostate cancer
**Specific** **Cell Types**	Pheno- Imager	[9]	2019	Colorectal cancer	FAP-expressing CAFs in the invasive tumor front in stroma-high tumors
	Pheno- Imager	[28]	2021	Melanoma	Enrichment of innate immune cells, specific DC subset and STAT3 expression Stage IV with leptomeningeal disease
	Pheno- Imager	[29]	2020	Colorectal cancer	TAMs subsets in stromal and epithelial compartments
	MIBI-TOF	[17]	2020	cSCC	Specific keratinocyte population within the TME
	IMC	[30]	2022	Lung cancer	Enriched PD-L1^+^CD8^+^ T cell subset in NSCLC
	IMC	[31]	2021	Breast cancer	High p-eIF4E expression in tumor cells and change immune cell composition
	IMC	[32]	2021	Colorectal cancer	Elevated proliferating and cytotoxic CD8^+^ T cell subsets in hypermutated CRC
**(Disease)** **Comparisons**	Pheno- Imager	[33]	2017	Prostate cancer	P300 increase and SIRT2 decrease when comparing BPH, prostate cancer to metastatic disease
	Pheno- Imager	[34]	2018	Esophageal cancer	High Notch Intracellular Domain expression in ESCC compared to benign or reactive epithelium
	MIBI-TOF	[35]	2022	Breast cancer	Comparing fibroblast composition in healthy breast tissue, DCIS, and IBC.
	IMC	[31]	2021	Breast cancer	Comparing Immune cell composition before, during and after pregnancy
	IMC	[32]	2021	Colorectal cancer	Comparing Immune cell composition in DB-CRC and nDB-CRC
	IMC	[36]	2018	Prostate cancer	Comparing bone marrow, prostate, and metastatic tissue
	DSP	[26]	2019	Prostate cancer	Different (immune) cell compositions and signaling pathways when comparing the TME of lytic and blastic bone metastasis
	DSP	[37]	2020	Endocrine tumors	Comparing the TME of neuroendocrine tumors and neuroendocrine carcinomas
	DSP	[38]	2021	Glioblastoma	Comparing immune-oncology proteins in methylated and unmethylated isocitrate dehydrogenase wild-type glioblastoma
	DSP	[39]	2021	Breast cancer	Comparing immune cell profiles in luminal and basal-like breast cancer
	DSP	[40]	2021	Colorectal cancer	Comparing the TME after neoadjuvant chemotherapy alone or in combination with ICPI in CRC patients
**Cellular** **Interactions**	Pheno- Imager	[41]	2018	Lung cancer	Tumor–T cell interactions in tumor core and CD8^+^ T cell–T_reg_ cells associated with overall survival in NSCLC
	Pheno- Cycler	[42]	2022	Breast cancer	Increased interaction between CD4^+^ and CD8^+^ T cells after cPLA2 treatment in mice
**Novel** **Immune**	IMC	[23]	2021	Lung cancer	Identification of CD3^−^CD4^+^FOXP3^+^CD25^−^CD127^−^TNFα^+^IFNγ^−^TdT^+^ cells in NSCLC (SCC)
**Subtypes**	IMC	[43]	2020	Hodgkin lymphoma	CD4^+^LAG3^+^ T cells in MHC-II negative classic Hodgkin Lymphoma
	IMC	[44]	2019	Colon cancer	CD4^+^EpCAM^+^PD-L1^+^ T cells with upregulated p38-MAPK-MAPKAPK2 pathway

Abbreviations: BPH: benign prostatic hyperplasia, CD3: cluster of differentiation 3, CD4: cluster of differentiation 4, CD8: cluster of ddifferentiation 8, CD25: cluster of differentiation 25, CD127: cluster of differentiation 127, CAF: cancer-associated fibroblast, cPLA2: cytosolic phospholipase A2, CRC: colorectal cancer, cSCC: cutaneous squamous cell carcinoma, DB: durable benefit, DC: dendritic cell, DCIS: ductal carcinoma in situ, DSP: digital spatial analysis, EpCAM: epithelial cell adhesion molecule, ESCC: esophageal squamous cell carcinoma, FAP: fibroblast activation protein, FOXP3: forkhead box P3, ICPI: immune checkpoint inhibition, IBC: invasive breast cancer, IFNγ: interferon gamma, IMC: imaging mass cytometry, LAG3: lymphocyte activation gene 3, MAPK: mitogen-activated protein kinase, MAPKAPK2: mitogen-activated protein kinase-activated protein kinase 2, MHC-II: major histocompatibility complex II, MIBI-TOF: multiplexed ion beam imaging by time-of-flight, nDB: non-durable benefit, NSCLC: non-small cell lung cancer, OSCC: oral squamous cell carcinoma, PD-L1: programmed death-ligand 1, p-eIF4E: phospho-eukaryotic translation initiation factor 4E, SCC: squamous cell carcinoma, SIRT2: sirtuin 2, STAT3: signal transducer and activator of transcription 3, TAM: tumor-associated macrophages, TdT: terminal deoxynucleotidyl transferase, TME: tumor microenvironment, TNBC: triple-negative breast cancer, TNFα: tumor necrosis factor alpha, T_reg_: regulatory T cell.

**Table 3 cancers-14-03170-t003:** Summary table of multiplex imaging studies that describe clinical outcomes, treatment responses, biomarkers, and tumor classification and grading.

Category	Method	Ref	Year	Cancer Type	Described Observations
**Clinical** **Outcome**	Pheno- Imager	[41]	2018	Lung cancer	Tumor–T cell interactions in tumor core and CD8^+^ T cell: T_reg_ cell ratios were associated with overall survival in NSCLC
	Pheno- Imager	[45]	2021	Ovarian cancer	High ratios of CD8:FOXP3 and CD8: PD-L1 T cells ratios were associated with favorable overall survival in high-grade serous OC
	Pheno- Imager	[34]	2018	Esophageal cancer	High-notch intracellular domain-expressing ESCC tumors have a decreased overall survival rate
	Pheno- Imager	[46]	2017	Renal cell carcinoma	PD-1^+^LAG3^+^CD8^+^ T cells were associated with poorer 36 month overall survival and higher relapse risk
	Pheno- Imager	[47]	2015	Prostate cancer	Lowest quartile nuclear SBP1 expression levels were associated with a higher recurrence risk after radical prostatectomy
	Pheno- Imager	[48]	2022	Ovarian cancer	Increased PD-L1 macrophages, ICOS^+^ T_h_ > T_reg_ numbers post-therapy, and decreased proximity ICOS^+^ T_h_ to T_reg_ cells in high-grade serous OC
	Pheno- Cycler	[49]	2020	Colorectal cancer	Specific cellular neighborhoods are associated with overall survival
	MIBI-TOF	[21]	2018	Breast cancer	Compartmentalized tumors are associated with increased survival in TNBC
	MIBI-TOF	[35]	2022	Breast cancer	Progressors from DCIS to IBC had a thicker and continuous MEC layer
	IMC	[22]	2020	Breast cancer	Single-cell pathology grouping improved the prediction of overall survival in TNBC
	IMC	[25]	2021	Bladder cancer	Stem-like cell cancer cluster (ALDH^+^PD-L1^+^ER-β^−^) is associated with poor prognosis in MIBC
	IMC	[50]	2022	Breast cancer	Structures containing T_regs_ and exhausted T cells and structures enriched in granulocytes or APC are correlated with poor prognosis in ER^−^, but not ER^+^ breast cancer tumors
	IMC	[51]	2021	Ovarian cancer	LTS showed increased number of granzyme B^+^ CTLs and CD45RO^+^CD4^+^ T cells, and a reduction in tumor cells and endothelial cells in high–serous OC. Granzyme B^+^ CD8^+^ T cells and CD45RO CD4+ interactions were correlated with overall survival
	IMC	[52]	2019	Gastric cancer	Responding mFOLFOX-treated tumors showed higher platinum levels compared to non-responders
	IMC	[53]	2021	Lung cancer	Abundant Ebo CD8^+^ TILs are correlated with poor overall survival in NSCLC
	MALDI- MSI	[54]	2019	Breast cancer	Identified nine proteins associated with EGFR related to progression in TNBC
	MALDI- MSI	[55]	2021	Colorectal cancer	Different N-glycosylation patterns in TME to distinguish short- and long-term survivors
	MALDI- MSI	[56]	2016	Ovarian cancer	Different N-glycosylation patterns in TME to distinguish short- and long-term survivors
	DSP	[57]	2020	Breast cancer	High CD4 and ICOS expression in stroma and HLA-DR expression in stroma or epithelial compartment were associated with long-term disease-free survival in TNBC
	DSP	[58]	2019	Lung cancer	PD-L1 expression in the macrophage compartment was associated with progression-free survival and overall survival in NSCLC
	DSP	[59]	2020	Lung cancer	High CD4 and CD56 expression in the immune cell compartment was associated with overall survival, progression-free survival, and durable benefit in NSCLC
	DSP	[60]	2020	B-cell lymphoma	High LAG3 expression was associated with poorer progression-free survival and overall survival
	DSP	[61]	2020	Lung cancer	Expression of CD3, ICOS, and CD34 in the tumor compartment was associated with improved overall survival in NSCLC
	DSP	[62]	2018	Melanoma	Low CD3, B2M, and PD-L1 and low IFNγ signature were associated with relapse after adjuvant or neoadjuvant ipilimumab and nivolumab
**Treatment** **Response** **Human**	Pheno- Imager	[63]	2021	Head and neck SCC	CD3^+^ T cells and CD8^+^ T cells are increased post-treatment with cetuximab in responders, compared to pre-treatment
	Pheno- Imager	[64]	2017	Rectal cancer	Lower CD4: PD-L1, CD8: PD-L1, FOXP3: PD-L1 ratios in total regression compared to residual disease
	IMC, MALDI -MSI	[65]	2022	Pancreatic cancer	Gemcitabine metabolites induce γH2AX in KI67^+^ Phosphorylated-ERK^+^ and Phosphorylated-S6^+^ areas in pancreatic ductal adenocarcinoma
	IMC	[66]	2019	Breast cancer (HER2^+^)	Elevated ECD/ICD ratio in cytokeratin positive compartment had a lower number of 5-year reoccurrence after trastuzumab
	IMC	[67]	2021	Rectal cancer	Reduced T_reg_ cells and TAMs, higher CTL levels were associated with complete response
	IMC	[68]	2021	Gastro-esophageal	Comparing immune cell composition changes in ramucirumab/paclitaxel-responding patients with or without ICPI administration in gastro-esophageal adenocarcinoma
	DSP	[69]	2021	Hairy cell leukemia	Changes in CD8 expression and tumor burden are associated with a durable response to cladribine
	DSP	[58]	2019	Lung cancer	PD-L1 expression in the macrophage compartment was associated with immune therapy response in NSCLC
	DSP	[70]	2021	Head and neck SCC	Immune cell number and CD4, CD68, CD45, CD44, and CD66b were correlated with progressive disease during ICPI treatment
	DSP	[71]	2018	Melanoma	CD45RO, B2M, CD3, CD8, CD19, CD20, and Ki67 in the immune cell compartment were associated with ICPI response
**Treatment** **Response** **Mice**	Pheno- Imager	[72]	2022	HPV16^+^ solid tumors	Total macrophages, activated CTL, and number of activated T cells are higher in responders compared to non-responders to HPV16^+^ vaccine (ISA101) with nivolumab
	Pheno- Cycler	[42]	2022	Breast cancer	CD4^+^ and CD8^+^ T cell infiltration and interaction increased upon cPLA2 inhibitor treatment
	IMC	[73]	2020	Biliary tract cancer	Elevated CD8^+^ T cell numbers in small anti-PD-1 sensitive tumors
	IMC	[74]	2021	Lung cancer	Change in immune infiltrate upon KRAS inhibition in NSCLC
	IMC	[75]	2021	Breast cancer	ITT and IPT anti-CD40/PD-L1 NDES treatment increased immune subsets and anti-tumor responses in TNBC
**Tumor** **Classification** **& Grading**	Pheno- Imager	[34]	2018	Esophageal cancer	Higher notch intracellular domain expression is correlated with higher tumor stage and grade in ESCC
	MALDI- MSI	[76]	2022	Lung cancer	Classifies SCC or AD (NSCLC) based on glutamine or taurine in the TME
	MALDI- MSI	[77]	2019	Lung cancer	Spatial distribution of CK5/6, HSP27, and CK15 to classify SCC or AD (NSCLC)
	MALDI- MSI	[78]	2010	Breast cancer	Identification of cysteine-rich protein 1 in HER2^+^ breast cancer
	MALDI- MSI	[79]	2020	NSCLC	Elevated levels of collagenase type III can discriminate low-grade adenocarcinoma from healthy lung tissue
**Biomarker** **Discovery**	IMC	[80]	2021	Melanoma	B2M expression in tumor and stroma compartment is correlated with longer overall survival to anti-PD-1 therapy
	MALDI- MSI	[81]	2020	Lung cancer	Identification of neutrophil defensins in ICPI-responsive NSCLC patients
	MALDI- MSI	[54]	2019	Breast cancer	Identified nine proteins associated with EGFR related to progression in TNBC

Abbreviations: AD: Adenocarcinoma, ALDH: aldehyde dehydrogenase, B2M: beta-2 microglobulin, CD3: cluster of differentiation 3, CD4: cluster of differentiation 4, CD8: cluster of differentiation 8, CD19: cluster of differentiation 19, CD20: cluster of differentiation 20, CD34: cluster of differentiation 34, CD40: cluster of differentiation 40, CD44: cluster of differentiation 44, CD45: cluster of differentiation 45, CD56: cluster of differentiation 56, CD66b: cluster of differentiation 66b, CD68: Cluster of Differentiation 68, CK5/6: Cytokeratin 5/6, CK15: Cytokeratin 15, cPLA2: cytosolic phospholipase A2, CTL: cytotoxic T cells, DCIS: ductal carcinoma in situ, DSP: digital spatial analysis, Ebo: burned-out effector, ECD: extracellular domain, EGFR: epidermal growth factor receptor, ER: estrogen receptor, ER-β: estrogen receptor beta, ERK: extracellular signal-regulated kinase, ESCC: esophageal squamous cell carcinoma, FOXP3: forkhead box P3, γH2AX: gamma h2a histone family member X, HER2: human epidermal growth factor receptor 2, HLA-DR: human leukocyte antigen—DR isotype, HPV16: human papilloma virus 16, HSP27: heath shock protein 27, ICD: intracellular domain, ICPI: immune checkpoint inhibition, ICOS: inducible T cell costimulation, IBC: invasive breast cancer, IFNγ: interferon gamma, IMC: imaging mass cytometry, IPT: intraperitoneal treatment, ITT: intratumoral treatment, KRAS: Kirsten rat sarcoma viral oncogene homolog, LAG3: lymphocyte activation gene 3, LTS: long-term survivors, MALDI-MSI: matrix-assisted laser desorption ionization mass spectrometry imaging, MEC: myoepithelial E-cadherin, mFOLFOX: modified folinic acid, fluorouracil, and oxaliplatin, MIBC: muscle invasive bladder cancer, MIBI-TOF: multiplexed ion beam imaging by time-of-flight, NDES: nanofluidic drug-eluting seed, NSCLC: non-small cell lung cancer, OC: ovarian carcinoma, PD-1: programmed cell death protein 1, PD-L1: programmed death-ligand 1, SBP1: selenium binding protein 1, SCC: squamous cell carcinoma, TAM: tumor-associated macrophages, TIL: tumor-infiltrating lymphocyte, TME: tumor microenvironment, TNBC: triple-negative breast cancer, T_reg_: regulatory T cell.

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
