# Peer review of "Multiplex Tissue Imaging: Spatial Revelations in the Tumor Microenvironment"

_cancers, 2022, doi:10.3390/cancers14133170_

Round 1

Reviewer 1 Report

Comments and Suggestions for Authors

In this paper Stephanie van Dam et al have reviewed multiplex spatial analysis methods that have advanced cancer research tremendously. They have reviewed different multiplex spatial imaging methods, that include microscopy (PhenoImager, PhenoCycler), mass-spectrometry (Imaging Mass Cytometry (IMC), Multiplexed ion beam imaging by time-of- flight (MIBI), and matrix-Assisted Laser Desorption Ionization Mass Spectrometry Imaging (MALDI-MS)), or digital spatial profiling (DSP). They have review very well their differences, clinically relevant applications and technical details.

Review needs to address minor comments below before acceptance.  

Comments:

1.) Figure 1.  style can be improved. Why there is no tumor stage IV? Please refer to BioRender (https://biorender.com/) or similar.

2.) Table 1. In the column described observations please unify the text, while using brackets for immune or not, and to use immune cell composition or just immune composition,…

Abbreviations under the table should be added.

3.) Grammar and rephrasing some sentences throughout the text is needed.

4.)  In challenges section the Data Management Plan should be mentioned, problems with saving and processing the metadata produced.

Author Response

We would like to thank all 3 reviewers for their constructive comments, and we have adapted the manuscript accordingly. Please see the attachment, which includes a point-by-point response to your comments.

Reviewer 2 Report

The article aims to review the different multiplex imaging techniques and analysis methods to study tumors with their microenvironment, and their applications in research and in clinics.

The article is very interesting and covers well the subject. I would recommend it for publication, but before, there are some important clarifications and reorganizations that have to be done, to facilitate the reading and understanding, please.

Specific comments :

-line 93: I recommend adding a simple definition of what is Multiplex spatial imaging methods to begin the paragraph.

-"2. Discoveries within the TME ...": I would recommend inverting the order of the tables. Presenting, first, the different methods, before the studies would be clearer. 

Also, you should add a figure before the "Table 1" and "Table 2", so after presenting the techniques, with images of what we can observe with these techniques (to help the reader to visualize what kind of images can be produced).

Finally, "Table 1" and "Table 2" must be reorganized to have the same order of presentation of the techniques and associated results for each category following the order used in the article. That is to say, your part "2.1" is about "PhenoImager", so in "Table 1" "TME Heterogeneity", the first method should be "PhenoImager", to have a table that follows the logic of your article, as you did in "Table 3".

-lines 150-152: The sentence is a bit complex to understand, it needs to be rewritten, please.

-All paragraphs: To be clearer and easier to read, you should align your text in "Justify" mode, and add a carriage return when you talk about a different notion (or reference for sub-paragraphs). For instance, in the paragraph "2.1 PhenoImager", you present 3 different notions: IHC, Multiplex IF, and then PI. However, as written currently, it is a little bit confusing and difficult to follow you.

-"2.1.1. Cell-Cell Interaction ...": The link with PhenoImager is not clear nor in the title, or in the text (does "Vectra imager" an alternate name for PhenoImager ?). Maybe, you should add a small sentence in the previous paragraph "2.1" to introduce the two subparagraphs. 

-All paragraphs: All paragraphs should take into account the previous comment, please.

-Article: A question out of curiosity, please. Why did you choose to develop in the paragraphs, "these" references and put the others in "Table 1" and Table 2"? 

-Article: Must be added for each paragraph corresponding to a technique, figure representing schematically the principle of the technique. It would be a high benefit for the article, attract more readers, and highly improve the understanding.

-"3. Challenges ...": Very interesting paragraphs. 

Author Response

We would like to thank all 3 reviewers for their constructive comments, and we have adapted the manuscript accordingly. Please see the attachment for a  point-by-point response to your comments.

Reviewer 3 Report

Regarding the review you sent me, it is fully discussing the new multiplex technology for evaluation of tumor microenvironment and their role in diagnosis, spatial distribution and potential detection of new biomarkers. I think the review is interesting in because it collect a wide spread of data in this unique field of study which supported by most recent references in this field. i have the following comments and suggestions2.3 more details regarding MIPI-TOF only one ref and no examples 3.1 no references line 590-600Line 637-639 more details missed.

Table 3 lacking ref

Line 649 = Naroset ???

In challenges (line 586-587) the authors mentioned many challenges and give a specific heading to only cell segmentation and tissue organization and cell localization.

Another point which should be addressed in the challenge related to the interobserver variability raised by the visual inspection which may interfere with the statistical power.

Along the entire review, the reviewer describes the main procedures in each imaging technique. it would be very interesting if the authors figure out the main steps that can be used in each technique in one figure. 

overall, the review is interisting and can be accepted after minor corrections

Author Response

(The authors gave the same response as above.)

Round 2

Reviewer 2 Report

I thank the authors for their answers, and for taking into account my suggestions.

The article is much clearer and easier to read and understand.

I recommend it for publication.

Here is a last very minor error to correct, please:

"Table 2 - Specific cell types": MIBI is placed after IMC.